# Identification of the needs of children with neurodisability and their families at different stages of development: A qualitative study protocol

**Patricia Roldán-Pérez, Marta San Miguel-Pagola, Víctor Doménech-García, Pablo Bellosta-López** [ID]*, **Almudena Buesa-Estéllez** [ID]

Universidad San Jorge, Campus Universitario, Villanueva de Gállego, Zaragoza, Spain

* pbellosta@usj.es

## Abstract

### Background

Within the field of childhood neurodisability, the tendency in the study of needs has been to categorize them based on ability (motor, verbal, cognitive). However, current perspectives such as F-words, family-centered practices, or the principles of family empowerment, lead the researcher to ask: What are these needs according to the stage of development?

### Methods and analysis

A descriptive qualitative study will be carried out. Several methods will be followed to ensure the reliability and validity of the results, and the Standards for Reporting Qualitative Research and the Consolidated criteria for reporting qualitative research checklists will also be used to guide the project. Data collection is sought from three main sources: Focus groups (detection of needs), a survey to collect sociodemographic and clinical data necessary to obtain an overview of the context of the participants, and a survey to find out the level of satisfaction with this initiative.

### Discussion

The results expected to be obtained after this study will respond to the main needs of families with childhood neurodisability, based on age groups and covering the whole territory of the Spanish population. Thanks to these detected needs, it will be possible to design future lines of work to improve the design of family-centered practices and increase the empowerment of families. The intention is to detect needs by stage of development, which can be categorized within the F-words framework, showing families and professionals a clear picture of the needs of this population.

relevant data from this study will be made available upon study completion.

**Funding:** This work was supported by "Ayudas a Proyectos Internos de Investigación Universidad San Jorge curso 2022-2023" grant number PI 22-23-029. The funders did not and will not have a role in study design, data collection and analysis, decision to publish, or preparation of the manuscript.

**Competing interests:** The authors have declared that no competing interests exist.

## Introduction

Neurodisability is an umbrella term attributed to brain or neuromuscular system impairments characterized by functional limitations and difficulties with movement, cognition, hearing and vision, communication, emotion, and behavior [1]. Traditionally, medical efforts have focused on diagnosing and treating chronic developmental disorders that cause neurodisability (e.g., intellectual disability, Down syndrome, cerebral palsy, spina bifida, and autism spectrum disorder [1]) from a biomedical perspective, with less attention paid to the functional capacities of individuals and their contextual determinants [2]. However, these aspects are important, especially in the management of childhood neurodisability [3], as parents are the main support providers for children with neurodisabilities at all stages of development [4]. Furthermore, each of these stages is unique and different in functioning, behavior, and context for each family unit [5].

Family-centered care is considered the best practice when providing services to children with neurodisability [6] and is associated with better outcomes based on meaningful goals set by children with neurodisabilities and their families. An essential aspect of family-centered care is the provision of resources, support, and services in response to family-identified needs and priorities [7]. However, families face significant challenges in identifying their personal, social, service-related, disability-related, and financial needs. In turn, unclear families' needs make children care even more challenging, as it might worsen pre-existing barriers such as the lack of proper access to information and environmental support [8].

The International Classification of Functioning, Disability and Health (ICF) and subsequent ICF-IA (childhood and adolescent version) [9] recognize the importance of self-care and environmental factors that facilitate comprehensive transition planning and service delivery for children and young people with chronic health conditions [10]. Since 2012, a conceptual framework for integrating ICF concepts into a model of applicability to both families and service providers, the "F-words; function, fitness, family, friends, fun, and future", has been developed [11]. This framework allows generalizing the natural terms of ICF into areas of development of children with disability, emphasizing their capabilities beyond their deficits, limitations, and restrictions in participation.

Therefore, the F-word "Function" relates to the activity of the ICF category, which describes activities in which an individual can participate, such as usual activities of daily living. "Fitness" is based on the ICF category of body structure and function, with "Family" being an essential element of children's environmental factors. "Friends" and "Fun" share components of the ICF relating to participation and personal factors, respectively. Finally, "Future" was included in the F-words to highlight the impact of a child's current condition on their future life, as well as "development" with its meaning relating to the future [11]. These terms are considered the six vital aspects for the holistic development of children with neurodisabilities and their highest level of well-being [12].

The literature closely links both family-centered practices and the F-words framework with the term's empowerment of the child and his or her family unit [12–14]. The term "family empowerment" allows families to have appropriate support that fosters involvement, knowledge, and a sense of management without generating stress or unwanted responsibilities [15].

For the design of actions that promote such empowerment, different studies have investigated the needs of families in different contexts, whether to understand the profile of families, the provision of necessary services, or financial and community support [16–19]. These studies have considered the level of gross motor function or the level of communication to categorize the needs. However, none of these studies has delved into the needs of children with neurodisability and their families within the framework of the F-words in the different stages of

development, not only establishing the categories according to motor and/or verbal ability but according to the vital stage of development of each child. Investigating families´ needs through the F-words at each stage of the children development at once is an innovative approach that can help (i) providing families with more accurate and tailored information regarding e.g., prognosis or severity, during early diagnostic process [20], which in turn could make families to cooperate more actively with the situation [21], (ii) observing potential differences across development stages and (iii) reducing the mismatch between families´ needs, healthcare services and policy-making [8].

It is therefore necessary to give a voice to children with neurodisability and their families through the lens of ICF and F-words at different stages of development. Thus, they can identify and express their experience regarding vital aspects of their daily lives that interfere with the holistic development of children in their different stages.

### Research objectives

The overall aim will be to identify and categorize the needs reflected by children with neurodisability and their families according to their stage of development. The specific aims will be:

1. To describe the perceptions of the real needs of the target population (children with neurodisability and their families).

2. To assess whether the needs detected are susceptible to being categorized within the conceptual framework of the ICF through the resource provided by the F-words.

3. To describe the sociodemographic and clinical characteristics of the different family units in a similar context.

4. To evaluate the level of satisfaction of participating in a project analysing their subjective experience.

## Materials and methods

### Study design

A descriptive qualitative study will be carried out [22]. Several methods will be followed to ensure the reliability and validity of the results, and the Standards for Reporting Qualitative Research (SRQR) and the Consolidated criteria for reporting qualitative research (COREQ) checklists will also be used to guide the project [23,24].

First-hand information will be collected on the participants' experience of a phenomenon, but the characteristics of the design will be different from those of a purely phenomenological design [25]. The resulting "lived experience" will be the perception, reflection, and awareness of situations or actions that have happened to the subject within a given geographical, social, and cultural environment. It is considered that this lived experience is always given meaning by the subject who has lived it [26].

When a much larger population is to be studied, with fewer resources or less time available, the design corresponds more to a case study, where multiple related topics can be studied using focus groups [27].

### Population

The study population will be families with at least one child with neurodisability who signed the informed consent to participate in the study and focus group interviews. The inclusion

criteria will comprise (i) fathers, mothers, or legal guardians of children with neurodisability at pediatric age (0–18 years); and (ii) children aged 12 to 18 years who will wish to participate in the focus groups with their fathers, mothers or legal guardians. The exclusion criteria will comprise (i) having a comprehension or oral communication problem in Spanish on the part of any of the subjects described in the inclusion criteria (fathers, mothers, guardians, or minors between 12 and 18 years of age with neurodisability); and (ii) lack of availability of a suitable electronic device for carrying out the focus groups online.

## Sample size

As there is no formula for sample size calculation in qualitative designs, the principle of theoretical data saturation will be used to finalize data collection [28]. In other words, the sample size will be large enough to obtain a variety of perspectives and small enough not to become cluttered or fragmented [29]. According to the literature, each focus group should have between 6 and 12 participants [30]. Data collection conducting focus groups will end when no new or relevant data emerge. Furthermore, it will be conditioned by the selection criteria, with at least one group representing all genders for each age range. They will be organized into age groups; 0–3 (early childhood), 3–6 (infancy), 6–12 (pre-adolescence), 12–18 years (adolescence).

## Contact and recruitment

To recruit families, a specific collaboration agreement has been created with the Spanish Society of Paediatric Physiotherapy (SEFIP), which will collaborate with the principal investigator (PRP) in disseminating the initiative to children and families who meet the inclusion criteria. An information poster and the participant information sheet will be developed for this recruitment. After contacting the principal investigator, reading the information (which includes instructions on the F-words framework), and asking any questions, potential participants will sign the informed consent form and complete the sociodemographic and clinical data survey to be able to participate in the focus groups. Participants will have to sign the informed consent form and return it in a legible format (scanned or photographed) via email.

The principal investigator will recruit and select participants. To select the families of children with neurodisability, theoretical or in-depth sampling will be used in a consecutive and non-probabilistic manner [31]. That is, selecting participants whose experiences align with the needs to be identified in the study's objectives. This sampling method will aim to gain a deeper understanding of aspects that may arise during the study.

## Data collection and security measures

Data is sought from three main sources: (i) Focus groups (detection of needs), (ii) a survey to collect sociodemographic and clinical data necessary to obtain an overview of the context of the participants, and (iii) a survey to find out the level of satisfaction with this activity.

**Focus groups.**   Focus groups are a suitable option for research and evaluation in the field of health [32], especially for those topics in which it is necessary to incorporate the perspective of patients, users, or the population at which the actions of health professionals are aimed [33]. Therefore, online focus groups will be conducted via Microsoft TEAMS, according to age ranges (0–3, 3–6, 6–12, 12–18 years).

Participants will be asked to express their most common doubts to identify their specific needs in the field of neurodisability following the contents of the F-words (i.e., function, fitness, family, friends, fun, and future). The list of questions to be developed is specified in Table 1.

**Table 1. Questions for online focus groups of families of children with neurodisability.**

They will be organized in age groups (0–3, 3–6, 6–12, 12–18 years) of children with neurodisability. Families and children will have a space to express their needs, addressing all aspects that underlie the F-words (function, fitness, family, friends, fun, and future). Children ages 12 to 18 will be invited to participate in the focus groups with their parents or legal tutors.
Each of the focus groups will begin with a round of introductions, so that participants can get situated among themselves.

**The first two questions will be:**
  • *As a family unit, what needs do you have concerning your child with pathology*?
  • *If you think about the stage of development in which your child is (early childhood, infancy, pre-adolescence, adolescence), in which of the following areas do you detect more needs; Function, Physical form, Family, Friends, Fun, Future*?

**The third open-ended question will be:**
  • *Focusing on your area of greatest concern, what are your questions and needs in this regard*?

In addition, other needs will be asked specifically:
  • *What other needs do you consider important, but not a priority now*?

At least one focus group will be carried out for each age range. Nevertheless, a second group would be carried out with participants recruited by theoretical sampling if it is necessary to go into some aspects in greater depth.

In the case there is more than one sibling with pathology in different age groups in the same family, they will be able to participate in more than one focus group according to the number of participants in the group (without exceeding 12, so that it does not become fragmented).

The focus groups will be recorded in their entirety, always after obtaining the permission of each participant to record it in the informed consent. In addition, observations and key data will be collected in the researchers' field notebooks by means of notes. After transcribing the information obtained in the groups into Office documents, the recording will be securely deleted. At the same time, a numerical code will be assigned to name each participant so that any data that could identify the participant disappears, thus ensuring their anonymity.

**Sociodemographic and clinical data.** To obtain information about the context of the participants in the focus groups, families will be asked for some sociodemographic data by means of a survey in Microsoft Forms, such as age, sex, cohabitation, and employment situation. Additionally, information regarding the child's age, diagnosis, type of schooling, therapies usually received, and whether the child requires orthoses, positioning systems, and mobility aids, among other things, will be collected (Appendix in S1 Appendix).

**Satisfaction survey.** The survey will be carried out through Microsoft Forms and sent to the households once their participation in the project is completed. The estimated time for the survey to be completed by the subjects who participated in the focus groups will be 20–25 minutes (Appendix in S2 Appendix).

**Face validity of the sociodemographic and clinical data & satisfaction surveys.** The surveys have been reviewed for clarity and comprehensiveness by a group of 4 parents of children with neurodisability and 4 therapists (occupational therapists and physical therapists).

## Data analysis

**Data analysis procedure.** Systematic Text Condensation will be used to analyze the patient focus groups [24]. This analysis procedure consists of an inductive thematic analysis which is not based on a previous theoretical framework [34]. In this analysis procedure, there are 4 steps that every phenomenological study should have to carry out its thematic analysis: (i) read to get a general idea of identifying preliminary themes to organize the data; (ii) identify

and classify the Units of Meaning (UoS), describing the grouping and distinguishing some codes from others; (iii) each grouping forms a unit of analysis, Common Meaning Groups (CSG) for further abstraction by grouping aspects that represent the thematic content; and (iv) synthesize the content of the condensation, creating a story based on the data reflecting content and meaning.

**Coding, grouping by units of meaning, identification, and development of themes and sub-themes.** The purpose of coding is to allow the researcher to simplify and centralize some specific features of the data, which will subsequently help in the phase of detection of themes and sub-themes [35]. Specifically, a list will be made of all UoS, identified or produced during the initial analysis. Then, repeated clusters will be eliminated by assigning a numerical code to each possible theme and sub-theme. Finally, all material obtained from the groups and participants' responses will be coded [35].

In the initial stages of coding and analysis, the level of abstraction of the data will be more descriptive. Progressively, the complexity and level of abstraction will increase to identify the essence. Thus, the analysis in the last phases will be more interpretative. In this thematic analysis proposal, a fully inductive analysis will be carried out, in which themes will emerge after the groupings of Units of meaning and common Groups of Meaning. Measure to ICF linkage, Measure to (Brief or Comprehensive) Core Set Absolute Linkage, Measure to (Brief or Comprehensive) Core Set Unique Linkage, Core Set Representation, and Core Set Unique Disability Representation indicators will not be used during the analysis but will be taken into account to relate concepts in the discussion after analysis of the results.

No software will be used to carry out the analysis. However, Microsoft Teams will be used to obtain immediate transcriptions, and an Office 365 application will be selected to create concept maps to help express the results.

**Methodological rigour.** In order to control the rigour and quality of the qualitative design, several procedures will be followed (Table 2) [36–47].

On the one hand, to ensure reliability: (i) there will be an inspection of the transcripts, following a previously established correct protocol in which the coding of specific parts is checked; (ii) there will be communication between team members; and (iii) there will be an attempt to present well-defined and described themes.

On the other hand, to maintain validity: first, the interviewer will try to avoid being "politically correct" during data collection with the informants, presenting and informing them of the study appropriately, favoring a relationship of trust and a relaxed and comfortable atmosphere. Besides, when asking questions, the interviewer will seek, with tact and respect, to go deeper to obtain answers (indirect questions, asking for examples, and relating them to the context). Second, the preliminary results will be checked with one of the informants, which also helps to maintain contact. These preliminary results will be identified, giving voice to the participants in this specific context and introducing strategies for change and improvement of clinical practice. Third, the research team will conduct a triangulation process using quantitative research support to discuss the results obtained during data analysis and peer checking. Finally, concerning the researcher on their own, the bracketing process will be carried out, incorporating the reflection process beforehand to add this more interpretative part at the end of the descriptive results.

**Role of the researcher: Bracketing and positioning of the researcher.** Before starting the data collection, it is essential to describe the role of the researchers due to (i) researchers are persons involved in the data collection; (ii) this data collection is not aseptic, which forces the researcher to interact with the participants and their social context; and (iii) the prior knowledge of the researcher may condition the way the researcher approaches the study phenomenon [48].

**Table 2. Methodological rigour.**

| CRITERIA | DESCRIPTION | TECHNIQUE USED | |
|---|---|---|---|
| **Credibility** | Confidence in the 'truth' of the findings. | Research team triangulation *(Denzin, 1978, Atkinson & Delamont, 2008)* | |
| | | Member checking *(Davis & Lachlan, 2017)* | |
| **Transferability** | Showing that the findings have applicablity in other contexts. *(Rose & Jonhson, 2020)* | Inspection of transcripts | |
| | | Following a previously established correct protocol | |
| | | Review the coding of specific parts | |
| | | Communication between team members | |
| | | Presentation of well-defined and described topics | |
| **Dependability** | Confidence in the 'truth' of the findings. *(Saldaña, 2013, Palacios-Ceña, 2014)* | Avoiding superficial coding | |
| | | Avoiding the researcher's interpretations | |
| | | Adding negative cases *(Maxwell, 2013)* | |
| | | External auditor *(Rose & Jonhson, 2020)* | |
| **Confirmability** | A degree of neutraility or the extent to which the findings of a study are shaped by the respondents and not researcher bias, motivation, or interest. *(Klem et al, 2021)* | Researcher herself/ himself | Bracketing *(Bergen, 2013)* |
| | | Participants | Avoid social desirability bias *(Bergen & Labonté 2020)* |
| | | | Member checking *(Davis & Lachlan, 2017)* |
| | | Research Team | Triangulation and Crystallization *(Richardson, 1997)* |
| | | | Peer debriefing |
| | | | Catalytic validity |

To describe all the elements that can condition the researcher during the study, bracketing makes it possible to "retain" or "sustain" the researcher's prior knowledge, to be able to go deeper into the fieldwork without preconceived ideas, and to obtain first-hand information without prior interpretations. Of course, this process of bracketing or " put in brackets " prior knowledge does not avoid the total influence of the researcher. However, it does contemplate its inclusion within the research protocol to consider it as another source of information within the study that can be reintegrated in later phases.

In this study, bracketing will be established in two ways. First, following Malterud's indications [49], describing the following elements: previous theoretical framework, research question, selected participants, beliefs about neurodisability, professional experience, and motivation guiding the research development. Second, the participants will be asked open questions to not condition or guide their answers, favoring dialogue within the group.

A researcher of the group (ABE, Physiotherapist with more than 14 years' experience in neurorehabilitation and qualitative interviews) will be in charge of conducting the focus groups, transcribing the content, and making the preliminary analysis of the data, later completed by the vision of the rest of the team members.

## Ethics and confidentiality

The study protocol has been prepared in accordance with the Helsinki Declaration and accepted by the San Jorge University Ethics Committee on 27 February 2023, with the number 37/1/22-23. Before the start of the project, specific information documents will be developed, which will be made available to families likely to participate in the study and show their interest. All participants will receive verbal and written information on the nature of the study and

will be able to consult any doubts related to it. Every participant willing to take part in the study will complete the informed consent form.

All data will be stored in line with General Data Protection Regulation (GDPR) guidance. The recordings from the focus groups using Microsoft Teams will be downloaded from the Stream (from which they will be deleted). They will be recorded on the researcher's computer in charge of the qualitative analysis, whose device is password protected with the university's own credentials. A review of the meeting will be carried out to check the transcript and add, if necessary, field notes on the behavior of the participants in the focus group. When the final transcript is available to work with, the recording will be securely deleted. The scanned informed consent documents, as well as the answers to the surveys in which the sociodemographic and clinical data of the participants will be collected, will be saved in a secure online folder of the university account to which only the staff of the research team will have access. All the computers of the researchers involved will be password protected.

## Dissemination

Any deviations from the protocol will be disclosed and justified when the results are published. Regardless of the outcome, the findings will be made public through peer-reviewed publications in open-access journals and important international conferences in the fields of health and behavioral sciences, pediatrics, and others.

## Discussion

The expected results will identify the main needs of families with childhood disabilities, categorized by age groups, covering the entire territory of the Spanish population. Thanks to these detected needs, it will be possible to devise future lines of work to increase the empowerment of families through the provision of resources. Previous investigations related to empowerment have considered a specific diagnosis (e.g., cerebral palsy), the level of gross motor function, or the level of communication without focusing on the vital stage of development of each child with different abilities [50,51]. Therefore, the novelty of our protocol lies in the grouping approach of the results, which focuses on the child's development stage.

Given the previous literature, it is expected that the main needs detected in the first age groups, which cover the childhood of children with disabilities, will show issues such as the acceptance of the child's condition by the parents. For example, the study by Fernández-Alcántara reflects the feeling of loss in parents based on grounded theory [52]. Other studies highlight the need for adaptations for activities of daily living, the most prominent being self-care and mobility [53]. Contextual factors that act as a barrier are another major concern for parents of children at early ages [54,55]. The lack of possibilities, the added difficulty presented in less familiar environments for children, and the attitude of others toward disability determine some of the obstacles faced by families in their day-to-day lives [54]. In these cases, while sharing results according to age group, the studies focus on the perceptions of parents of children with cerebral palsy [50]. Likewise, there are innovative studies that propose training programs for parents of children with neurodisabilities (0–6 years of age) that aim to provide information, support and preparation tools for the future and to get their opinion on the matter [56]; however, have not yet been applied to other developmental stages.

School is an important cross-cutting issue between childhood and adolescence, also included in previous literature. An important need of families is that their children are successfully enrolled in school. A great deal of teamwork is needed from parents, school professionals, and health professionals around the child, facilitating the promotion of an open and positive school culture built on inclusive problem-solving practices [57]. Empowering families by

favoring reflection on the usefulness of implementing autonomy-promoting strategies with their child and encouraging active participation in the educational engagement of children with disabilities can be key to facilitating school inclusion [58]. Therefore, describing in detail the specific needs of the children regarding "F-words" can contribute to creating better-tailored plans for successful school enrollment.

Regarding late childhood, preadolescence, and adolescence, other studies found focus on the transition to adulthood, in which people with disabilities (especially with cerebral palsy) are clear about their goals, what happiness is for them, and what they need to achieve it [59] (related to the F-words *Fun)*. The main barriers they face to their full development are intellectual disability and communicative problems [60], forcing them to live with their parents and not being able to work, unlike their peers without pathology (related to the F-words *Function* and *Family)*. Also, the existence of concomitant pathologies such as seizure disorders, asthma, obesity, urinary incontinence, and reflux disorders will condition their future prospects [61]. Therefore, knowing the overall needs map based on the "F words" will again provide a framework for generating resources to help plan for adulthood in all dimensions.

Responding to another of our objectives, developing this study will provide a global vision of the needs of children with neurodisabilities in the Spanish population. Although the data will be collected qualitatively, they will be completed with epidemiological information [62].

## Supporting information

**S1 Appendix. Satisfaction survey.**
(PDF)

**S2 Appendix. Sociodemographic and clinical data survey.**
(PDF)

## Acknowledgments

We thank the Spanish Society of Paediatric Physiotherapy (SEFIP) for their willingness to help us with recruitment. We also appreciate their dissemination among family associations and professionals linked to the area of care and research.

## Author Contributions

**Conceptualization:** Patricia Roldán-Pérez, Marta San Miguel-Pagola, Víctor Doménech-García, Pablo Bellosta-López, Almudena Buesa-Estéllez.

**Funding acquisition:** Patricia Roldán-Pérez.

**Methodology:** Almudena Buesa-Estéllez.

**Writing – original draft:** Patricia Roldán-Pérez, Pablo Bellosta-López, Almudena Buesa-Estéllez.

**Writing – review & editing:** Marta San Miguel-Pagola, Víctor Doménech-García.

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
