## [Decision Letter · Decision Letter 0]

10 Jul 2023

PONE-D-23-10164Identification of the needs of children with neuromotor pathology and their families at different stages of development: a qualitative study protocolPLOS ONE

Dear Dr. Bellosta-López,

Thank you for submitting your manuscript to PLOS ONE. After careful consideration, we feel that it has merit but does not fully meet PLOS ONE’s publication criteria as it currently stands. Therefore, we invite you to submit a revised version of the manuscript that addresses the points raised during the review process.

We look forward to receiving your revised manuscript.

Kind regards,

Renato S. Melo, PhD

Academic Editor

PLOS ONE

Journal Requirements:

**Additional Editor Comments:**

Dear authors, the reviewers have completed the work of reviewing your manuscript and have decided that the present study still needs some adjustments before it can be considered for publication. Therefore, the decision of the manuscript should be: Revise (Major revision).

Reviewers' comments:

Reviewer's Responses to Questions

**Comments to the Author**

1. Does the manuscript provide a valid rationale for the proposed study, with clearly identified and justified research questions?

Reviewer #1: Yes

Reviewer #2: Partly

2. Is the protocol technically sound and planned in a manner that will lead to a meaningful outcome and allow testing the stated hypotheses?

Reviewer #1: Yes

Reviewer #2: Partly

3. Is the methodology feasible and described in sufficient detail to allow the work to be replicable?

Reviewer #1: Yes

Reviewer #2: Yes

4. Have the authors described where all data underlying the findings will be made available when the study is complete?

Reviewer #1: Yes

Reviewer #2: Yes

5. Is the manuscript presented in an intelligible fashion and written in standard English?

Reviewer #1: Yes

Reviewer #2: Yes

6. Review Comments to the Author

You may also provide optional suggestions and comments to authors that they might find helpful in planning their study.

Reviewer #1: Thank you for the opportunity to review the manuscript. Indeed, the topic is relevant. However, I just have one question. In the inclusion criteria. Neuromotor pathology at pediatric age... what are they in fact?

Reviewer #2: I congratulate the authors for the effort in promoting current recommendations for research with children that have disabilities and their families. However, I believe the manuscript quality should be improved. I provide some suggestions next.

Overall: I feel that the authors could improve the study rationale by justifying why considering the classification levels is not sufficient (Perhaps at younger ages a prognosis is not clear? ) and why differences are expected across ages.

Also, perhaps not many studies have considered different developmental stages at once, but many studies have looked into specific age-ranges. A comprehensive literature overview is lacking, as well as a clear gap to be addressed.

Examples:

Byrne, R., Duncan, A., Pickar, T., Burkhardt, S., Boyd, R. N., Neel, M. L., & Maitre, N. L. (2019). Comparing parent and provider priorities in discussions of early detection and intervention for infants with and at risk of cerebral palsy. Child: Care, Health and Development, 45(6), 799-807.

Graungaard, A. H., & Skov, L. (2007). Why do we need a diagnosis? A qualitative study of parents’ experiences, coping and needs, when the newborn child is severely disabled. Child: care, health and development, 33(3), 296-307.

Resch, J. A., Mireles, G., Benz, M. R., Grenwelge, C., Peterson, R., & Zhang, D. (2010). Giving parents a voice: A qualitative study of the challenges experienced by parents of children with disabilities. Rehabilitation psychology, 55(2), 139.

page 4

When listing the F-words and respective ICF components, the authors missed the F-word Function.

Objectives:

2- Even though they are related, ICF components and F-words are not synonyms. Please adjust accordingly.

2 and 3- I do not see how these objectives fit the study rationale. Please clarify

Methods:

Please add information about study registration if applicable.

Procedures: Will participants receive any previous instructions on the f-words framework? If so, how they will be provided?

Data analyses: It is not clear if the authors will follow methods as recommended by the WHO for linking concepts to ICF (e.g., linking rules).

7. PLOS authors have the option to publish the peer review history of their article (what does this mean?). If published, this will include your full peer review and any attached files.

Reviewer #1: No

Reviewer #2: **Yes: **Ana Carolina de Campos

---

## [Author Response · Author response to Decision Letter 0]

20 Jul 2023

Thank you for the opportunity to revise our manuscript, “Identification of the needs of children with neuromotor pathology and their families at different stages of development: a qualitative study protocol” (PONE-D-23-10164). We are grateful for the time and effort that the editorial team and the reviewers have put into the response, which we feel has helped us improve the manuscript. Our point-by-point response has addressed all comments and suggestions, and we have changed the manuscripts accordingly. All responses and changes have been highlighted using a blue font for easier tracking in the tracked manuscript version. Furthermore, we attach a clean version of the revised manuscript.

We hope you and the reviewers find our responses to your satisfaction, and we look forward to hearing from you. 

Yours sincerely,

The authors

---

## [Decision Letter · Decision Letter 1]

23 Aug 2023

Identification of the needs of children with neurodisability and their families at different stages of development: a qualitative study protocol

PONE-D-23-10164R1

Dear Dr. Bellosta-López

We’re pleased to inform you that your manuscript has been judged scientifically suitable for publication and will be formally accepted for publication once it meets all outstanding technical requirements.

Kind regards,

Renato S. Melo, PhD

Academic Editor

PLOS ONE

Additional Editor Comments (optional):

Dear Dr. Bellosta-López,

Thank you for submitting your manuscript to PLOS ONE. After careful consideration, We are pleased to announce that your article entitled: Identification of the needs of children with neurodisability and their families at different stages of development: a qualitative study protocol, has been accepted for publication in PLoS One.

Reviewers' comments:

Reviewer's Responses to Questions

**Comments to the Author**

1. Does the manuscript provide a valid rationale for the proposed study, with clearly identified and justified research questions?

Reviewer #3: Yes

2. Is the protocol technically sound and planned in a manner that will lead to a meaningful outcome and allow testing the stated hypotheses?

Reviewer #3: Yes

3. Is the methodology feasible and described in sufficient detail to allow the work to be replicable?

Reviewer #3: Yes

4. Have the authors described where all data underlying the findings will be made available when the study is complete?

Reviewer #3: Yes

5. Is the manuscript presented in an intelligible fashion and written in standard English?

Reviewer #3: Yes

6. Review Comments to the Author

You may also provide optional suggestions and comments to authors that they might find helpful in planning their study.

Reviewer #3: The authors changed the manuscript as per the reviewers' recommendations. I believe that now the manuscript is clearer and more concise for the authors to understand.

7. PLOS authors have the option to publish the peer review history of their article (what does this mean?). If published, this will include your full peer review and any attached files.

Reviewer #3: No

---

## [Editor Report · Acceptance letter]

29 Aug 2023

PONE-D-23-10164R1 

Identification of the needs of children with neurodisability and their families at different stages of development: a qualitative study protocol 

Dear Dr. Bellosta-López:

I'm pleased to inform you that your manuscript has been deemed suitable for publication in PLOS ONE. Congratulations! Your manuscript is now with our production department. 

Kind regards, 

on behalf of

Dr. Renato S. Melo 

Academic Editor

PLOS ONE